Microbiology **Spectrum**
ठ | **Open Peer Review** | Antimicrobial Chemotherapy | Research Article

# Impact of the presence of a prosthetic implant and transition to oral stepdown therapy on relapse rates and mortality in uncomplicated *Staphylococcus aureus* bacteremia treated with 14 days of antibiotics: a retrospective cohort study

Damien Blez,[1,2] Luc Labarbe,[1] Patrick Grohs,[1] Jean-Luc Mainardi,[1,2,3] Jean-Philippe Barnier,[1,2] David Lebeaux,[1,2,4] Marie Dubert[1,2]

**ABSTRACT** The aim of this single-center, retrospective observational study was to evaluate the effects of having a prosthetic implant (PI) and of changing from intravenous to oral antibiotics (oral stepdown therapy [OST]) on the risk of relapse in patients with otherwise uncomplicated *Staphylococcus aureus* bacteremia (SAB) treated with antibiotics for 14 days. The primary outcome was the 90-day SAB relapse rate in patients with and without a PI. Secondary outcomes were 90-day mortality among patients with or without a PI and 90-day SAB relapse and mortality in patients who had OST. We included 188 consecutive patients with SAB without metastatic foci and with a planned antibiotic treatment duration of 14 days: 58 (31%) had a presumed uninfected PI, and 108 (57%) had OST. Four patients (2%) relapsed, and 25 patients (13%) died. Patients with a PI were more likely to have diagnostic tests performed. In the univariate analysis, the presence of a PI (odds ratio [OR] 7 [95% confidence interval {CI} 0.9–144.0]) and OST (OR 0.7 [95% CI 0.1–6.2]) were not associated with 90-day relapse. In the multivariable analysis, the presence of a PI (adjusted odds ratio [aOR] 1.3 [95% CI 0.5–3.7]) and OST (aOR 0.5 [95% CI 0.2–1.4]) were not predictive of 90-day mortality. In a setting where full diagnostic workup and close follow-up can be ensured, the presence of a PI and OST did not seem to be associated with an increase in 90-day mortality in patients with otherwise uncomplicated SAB. Although the relapse rate was low overall, there was a non-significant trend toward a higher risk of relapse in patients with a PI.

**IMPORTANCE** This retrospective study provides reassuring real-world data supporting a short 14-day treatment course for SAB in patients with PIs. In an era of increasing antimicrobial resistance worldwide, these retrospective findings support the perspective that not all PIs are systematically infected. Prolonged antibiotic therapy may therefore not be routinely needed if infection is excluded and thorough evaluation for dissemination performed, accompanied by close clinical and biological monitoring. Early transition to oral therapy in this context, which has been implemented in our institution for years, does not appear to be associated with a higher risk of therapeutic failure. These findings align with the most recent literature on the subject.

**KEYWORDS** *Staphylococcus aureus*, prosthesis, device, oral stepdown therapy, short-course antibiotic therapy

Address correspondence to Damien Blez, damien.blez@aphp.fr.

J.-L.M. is a member of the scientific committee of the BioAster Company and has participated in advisory boards with MSD Merck Sharp & Dohme AG.

*Staphylococcus aureus* bacteremia (SAB) is a severe disease, with 90-day mortality rates ranging from 15% to 50% despite the use of antibiotics (1–3). Antibiotic treatment for uncomplicated SAB is usually given for 14 days to prevent relapse (4, 5), but optimal management remains controversial as illustrated by the differences in treatment regimes

across countries (6). The 2010 Infectious Diseases Society of America guidelines on SAB included the presence of a prosthetic implant (PI) as a criterion for the definition of complicated SAB and recommended treatment with 4–6 weeks of parenteral antibiotic therapy in such patients (4). Indeed, the presence of a PI in a patient with SAB is known to be associated with an increased risk of relapse (7, 8). The hematogenous colonization of a prosthesis is, however, not constant in patients with SAB (9–12), and observational data suggest that low-risk patients with uninfected orthopedic prostheses or cardiac implanted electronic devices (CIEDs) could be managed without extended courses of antibiotic treatment (13).

An unnecessary, prolonged course of antibiotic therapy can have important consequences in terms of adverse effects, healthcare-related complications, and costs for society (14), especially if the intravenous (IV) route is used (15). As part of an antibiotic-sparing policy, the local guidelines at our hospital recommend 14 days of antibiotic therapy for uncomplicated SAB even if the patient has a PI, provided that metastatic foci and prosthesis infection have been excluded. We evaluated the risk of relapse associated with the presence of a presumed uninfected PI and with oral stepdown therapy (OST) in patients with uncomplicated SAB who were treated with antibiotics for 14 days.

## RESULTS

### Screening and inclusion of patients with uncomplicated SAB

Among 612 patients diagnosed with SAB between 2015 and 2021, 188 patients had uncomplicated SAB (Fig. 1), with a median follow-up duration of 6.7 months (interquartile range [IQR]: 1.4–19.6). Fifty-eight patients (31%) had a PI. The most common primary source of infection was catheter-related SAB (60%). The baseline characteristics of patients with and without a PI are shown in Table 1. Patients with a PI were older (74 vs 65 years, $P = 0.001$), less likely to have central-line access at the time of SAB (15 of 58 [26%] vs 60 of 130 [46%], $P = 0.01$), and less likely to have catheter-related SAB (25 of 58 [43%] vs 88 of 130 [68%], $P = 0.003$).

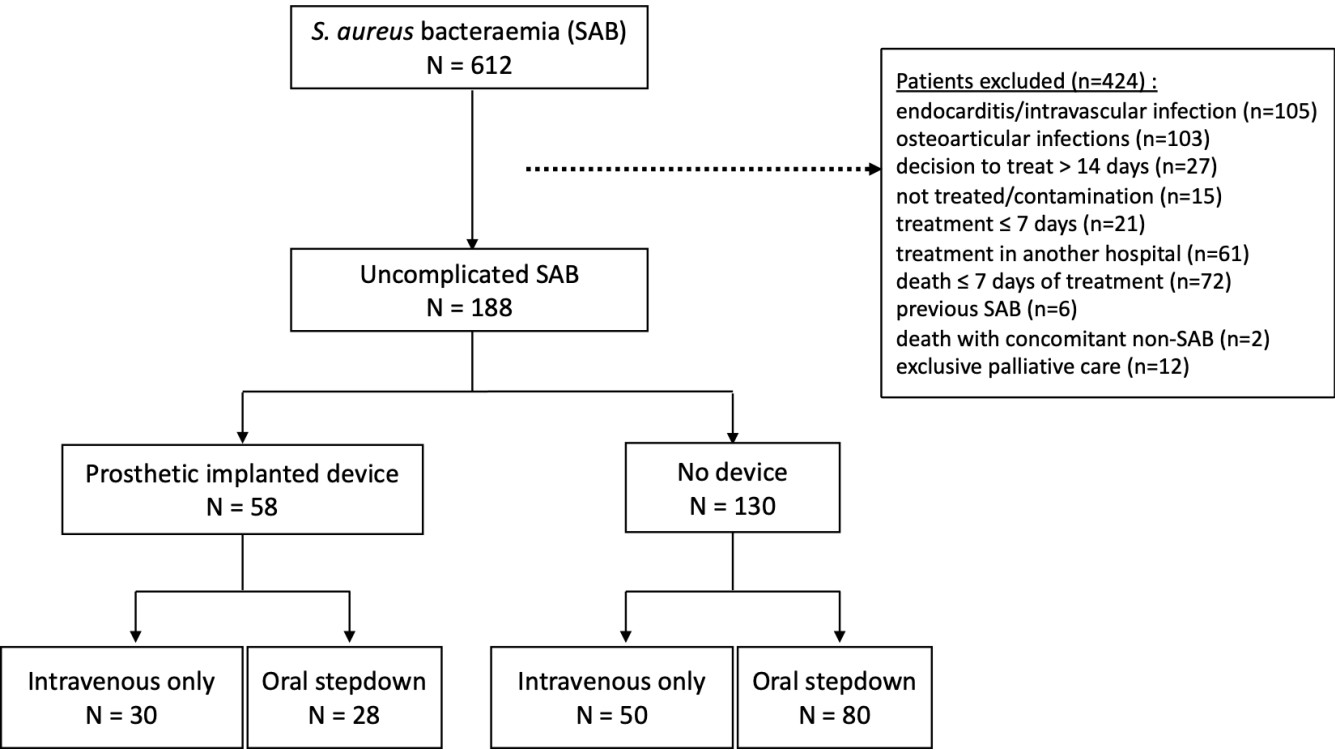

**FIG 1** Flowchart.

**TABLE 1** Baseline and clinical characteristics of the patients with SAB[d]

| | Prosthetic implant (%), n = 58 | No prosthesis (%), n = 130 | P value |
|---|---|---|---|
| Age (years), median (IQR) | 74 (65–85) | 65 (53–77) | 0.001 |
| Male | 42 (72) | 83 (64) | 0.6 |
| Patients at highest risk of IE[a] | 14 (24) | 0 | <0.001 |
| Central venous catheter at SAB onset | 15 (26) | 60 (46) | 0.01 |
| Orthopedic prosthesis | 22 (38) | 0 | <0.001 |
| Intravascular or cardiac prosthesis[b] | 42 (71) | 0 | <0.001 |
| Fever | 44 (76) | 109 (84) | 0.7 |
| SOFA score ≥2 | 19 (35) | 36 (28) | 0.6 |
| Nosocomial acquisition of SAB | 39 (67) | 88 (68) | 1.0 |
| MRSA SAB | 3 (5) | 13 (10) | 0.4 |
| Primary infectious focus (not exclusive) | | | |
| Catheter related | 25 (43) | 88 (68) | 0.003 |
| Skin and soft tissue | 7 (12) | 9 (7) | 0.4 |
| Pneumonia | 5 (9) | 8 (6) | 0.6 |
| Urinary tract infection | 7 (12) | 9 (7) | 0.4 |
| Unknown | 11 (19) | 12 (9) | 0.15 |
| Other[c] | 3 (5) | 7 (5) | 1.0 |
| Blood culture time to positivity | | | |
| <10 h | 17 (29) | 41 (32) | 0.9 |
| 10–15 h | 23 (40) | 39 (30) | 0.4 |
| >15 h | 14 (24) | 47 (36) | 0.3 |
| Unknown | 4 (7) | 3 (2) | 0.2 |

[a]Prosthetic valve, previous IE, untreated cyanotic CHD or treated CHD with residual shunt or valve regurgitation.
[b]Pacemaker (n = 17), implanted defibrillator (n = 8), prosthetic heart valve (n = 14), vascular graft (n = 15), and HeartMate (n = 1).
[c]Pacemaker/defibrillator lodge infection (n = 3), surgical site infection (n = 2), biliary tract infection (n = 2), peritonitis (n = 1), parotiditis (n = 1), and diabetic foot infection (n = 1).
[d]CHD, congenital heart disease; IE, infective endocarditis; IQR, interquartile range; MRSA, methicillin-resistant *Staphylococcus aureus*; SAB, *Staphylococcus aureus* bacteremia; SOFA, sequential organ failure assessment.

## SAB diagnostic workup

Transthoracic echocardiography (TTE) was performed in most patients (173 of 188 [92%]) and thoracoabdominopelvic computed tomography (CT) in 47 of 188 patients (25%), with no differences in those with or without a PI. Transesophageal echocardiography (TEE, 35 of 188 [19%]), fluorodeoxyglucose-positron emission tomography with CT (19 of 188 [10%]), and heart CT (4 of 188 [2%]) were performed in selected patients, more frequently in patients with a PI: 30% vs 14% ($P = 0.05$), 22% vs 4% ($P = 0.001$), and 7% vs 0% ($P = 0.01$), respectively (Table S1).

## SAB management

Intravenous antibiotic therapy consisted of β-lactams (cefazolin [85%], cloxacillin [13%], or amoxicillin when penicillin susceptible [2%]) in 156 of 172 (91%) patients with methicillin-susceptible SAB, either as initial treatment (n = 114) or after susceptibility testing while receiving vancomycin therapy (n = 42). OST was performed in 108 out of 188 patients (57%) after a median of 7 days (IQR: 5–9), mainly using clindamycin (46%) and cotrimoxazole (31%). The total duration of antibiotic treatment was 14 days (IQR: 14–16) for patients with a PI and 14 days (IQR: 14–15) for those without. In 21 out of 188 (11%) patients, SAB persisted for more than 72 h despite appropriate antibiotic treatment; this was more frequently seen in patients with a PI (19 vs 8%, $P = 0.05$) (Table S2).

## Primary and secondary outcomes

Overall, 4 of 188 (2%) patients relapsed after a 14-day (n = 3) and 16-day (n = 1) antibiotic therapy course, between 7 and 32 days after the end of antibiotic therapy (Table 2).

**TABLE 2** Primary and secondary outcomes at 90 days[b]

|  | Prosthetic implant (%), *n* = 58 | No prosthesis (%), *n* = 130 | *P* value |
|---|---|---|---|
| SAB relapse | 3 (5) | 1 (1) | 0.1 |
| All-cause mortality | 11 (19) | 14 (11) | 0.3 |
| SAB at the time of death | 1 (1) | 0 | –[c] |
| Missing blood culture results | 4 of 11 | 4 of 14 | – |
| Alive at 3 months | 40 (69) | 86 (66) | 0.9 |
| Follow-up (months [IQR]) | 6.6 (2.0–15.0) | 6.7 (1.1–25.8) | 0.9 |
| Loss to follow-up at day 30[a] | 5 of 47 (11) | 20 of 116 (17) | 0.5 |
| Loss to follow-up at day 90[a] | 7 of 47 (15) | 31 of 116 (27) | 0.6 |

[a]Deceased patients excluded.
[b]IQR, interquartile range.
[c]"–" indicates the absence of analysis.

Relapse occurred in 3 of 58 (5%) patients with a PI and in 1 of 130 (1%) patients without (*P* = 0.1) (Table 2; Tables S3 and S4). These four patients all had methicillin-susceptible SAB. The final diagnosis in these patients was unproven infective endocarditis (IE) (*n* = 2: one prosthetic valve IE and one pacemaker lead IE), infected psoas hematoma (*n* = 1), and ventilator-associated pneumonia (*n* = 1). One of these patients died from gastrointestinal bleeding 4 weeks after SAB relapse while still receiving antibiotic treatment and with repeatedly negative follow-up blood cultures. The other three patients were cured, with no recurrent SAB after a follow-up of more than 110 days.

In the univariate logistic regression, none of the variables studied was associated with relapse, including presence of a PI and OST (Table S3). Multivariable analysis could not be performed for factors predicting risk of relapse because of the small number of events (*n* = 4).

The all-cause 90-day mortality rate was 25 of 188 (13%): 11 of 58 (19%) patients with a PI died and 14 of 130 (11%) without (*P* = 0.25) (survival curve shown in Fig. S1). One patient with a PI died with unresolved SAB. Factors associated with increased 90-day mortality in univariate logistic regression are shown in Table 3. In the multivariable analysis, sequential organ failure assessment score (adjusted odds ratio [aOR] 3.54 [1.34–9.67]) and blood culture time to positivity of <10 h (aOR 4.00 [1.37–12.57]) were independently associated with increased 90-day mortality. Presence of a PI (aOR

**TABLE 3** Logistic regression for prediction of 90-day mortality in patients with SAB[b]

|  | Univariate analysis | | Multivariable analysis |
|---|---|---|---|
|  | OR (95% CI) | *P* value | aOR (95% CI) |
| Prosthetic implant | 1.9 (0.8–4.6) | 0.13 | 1.3 (0.5–3.7) |
| Oral stepdown therapy | 0.2 (0.1–0.6) | **0.002**[c] | 0.5 (0.2–1.4) |
| Age | 1.0 (0.98–1.0) | 0.55 | –[d] |
| SOFA ≥2 | 4.8 (2.0–12.0) | **0.0005** | **3.5 (1.3–9.7)** |
| MRSA | 0.9 (0.14–3.6) | 0.92 | – |
| Blood culture TTP <10 h | 2.3 (1.0–5.6) | **0.05** | **4.0 (1.4–12.6)** |
| SAB duration ≥72 h | 3.1 (1.0–8.7) | **0.04** | 2.26 (0.6–8.0) |
| Primary infectious focus |  |  |  |
| Catheter related | 0.3 (0.1–0.8) | **0.01** | 0.6 (0.2–2.4) |
| Primary cutaneous | 0.4 (0.02–2.2) | 0.40 | – |
| Pneumonia | 3.3 (0.8–11.0) | **0.07** | 4.71 (1.0–22.3) |
| Urinary tract infection | 0.9 (0.1–3.6) | 0.92 | – |
| Unknown | 4.6 (1.7–12.0) | **0.002** | 3.4 (0.8–14.6) |
| Other[a] | 0.7 (0.04–4.1) | 0.75 | – |

[a]Pacemaker/defibrillator lodge infection (*n* = 3), surgical site infection (*n* = 2), biliary tract infection (*n* = 2), peritonitis (*n* = 1), parotiditis (*n* = 1), and diabetic foot infection (*n* = 1).
[b]aOR, adjusted odds ratio; CI, confidence interval; MRSA, methicillin-resistant *Staphylococcus aureus*; OR, odds ratio; SAB, *Staphylococcus aureus* bacteremia; SOFA, sequential organ failure assessment; TTP, time to positivity.
[c]Boldface values for P < 0.1, included in the multivariable analysis.
[d]"–" indicates the absence of analysis.

1.34 [0.47–3.7]) and OST (aOR 0.47 [0.16–1.36]) had no impact on 90-mortality in the multivariable logistic regression model (Table 3).

## DISCUSSION

In this 6-year, single-center, retrospective cohort of 188 patients with uncomplicated SAB treated with a 14-day course of antibiotics, there was no significant difference in the 90-day relapse rate in patients with a presumed uninfected PI compared to those without a PI. Moreover, the presence of a PI and the use of OST were not independently associated with 90-day mortality in the multivariable analysis.

We observed a 90-day relapse rate of 2% and a 90-day all-cause mortality of 13%, which correspond to the lower ranges in published data (2.1%–20.3% for relapse rate [8] and 15%–50% for mortality rate [1–3]). This low treatment-failure rate despite short-course antibiotic treatment and presence of a PI may be explained by the careful selection of patients through systematic patient evaluation for infection and close follow-up by the antibiotic stewardship team at our institution (16). Nevertheless, it should be noted that the exclusion of patients who died early (before 7 days) may have led to an underestimation of overall mortality compared to published data. While some patients may relapse without a new positive blood culture, none of the patients in our cohort received additional antistaphylococcal treatment during follow-up (data not shown).

Although a precise definition of complicated SAB has not been established, most experts consider PI to be a component of "complicated SAB" (17). It has been reported that orthopedic prostheses are infected in 39%–41% of patients with SAB (18, 19). However, most diagnoses of prosthesis infection are made during the initial SAB episode. Among patients with a clinically uninfected orthopedic prosthesis at SAB presentation, only 4 of 50 (8%) (18) and 0 of 19 (19) developed subsequent prosthetic joint infection (PJI). Moreover, extended courses of treatment failed to prevent these infections in three out of four patients (18), and the fourth patient had a PJI 315 days after a 15-day course of cefazolin, which suggests reinfection rather than relapse.

Few studies have described SAB outcomes in patients with an uninfected cardiovascular prosthesis: in one study, 2 out of 30 patients with an uninfected CIED had an early SAB relapse after a short course of antibiotic treatment. None of these patients had had a TEE. One died, and the other received another short course of antibiotic treatment and had no relapse during a 4-month follow-up period (10). In another study (11), there was a 19% SAB relapse rate in 142 SAB patients with uninfected CIEDs. A duration of bacteremia of more than 1 day was strongly associated with relapse (35% vs 5%, $P < 0.0001$). This high relapse rate occurred despite a median antibiotic course of 28 days, raising the possibility of inadequate diagnosis and source control rather than inadequate antibiotic duration. Finally, in a retrospective study of patients with SAB and an uninfected prosthetic cardiac valve, 1 of 27 and 2 of 32 relapsed after treatment durations of less than 14 days or 14–27 days, respectively (20).

Of the four relapses in our study, one patient had an orthopedic prosthesis (and a prosthetic heart valve), which was not considered to be infected at the time of relapse, and three patients had a cardiovascular prosthesis, two of which were finally presumed to be infected, without morphological evidence.

Kaasch et al. observed no increased risk of mortality or relapse in 43 low-risk patients with a PI compared to 249 patients without a prosthesis (13). However, these authors excluded patients with prosthetic heart valves, vascular grafts, or HeartMate from the analysis.

Together, these data support the overall safety of a 14-day antibiotic course for patients with SAB who have a presumed uninfected prosthesis, provided an adequate diagnostic workup has been performed, and the clinical course and microbiological follow-up are suggestive of an uncomplicated infection.

Although parenteral antibiotic therapy has long been considered the standard treatment for SAB, recent international surveys have indicated that oral treatment is

increasingly considered a valuable alternative (6, 17). Indeed, OST has been reported to be efficacious in retrospective studies of patients with uncomplicated SAB (21–23) and in complicated SAB in intravenous drug users (24), but patients with a PI were mostly excluded from these studies or the types of prostheses were not specified. OST was also non-inferior to parenteral therapy in patients with endocarditis in a multicenter randomized trial, in which 27% of patients had a prosthetic valve and 87 patients had *S. aureus* endocarditis (25). Results from the *Staphylococcus aureus* Bacteremia Antibiotic Treatment Options (SABATO) trial also suggested non-inferiority of OST in a highly selected population of patients with uncomplicated SAB (26), but patients with a cardiovascular prosthesis and some orthopedic prostheses were excluded. A randomized clinical trial of early OST in patients with uncomplicated and complicated SABs is currently under way within the *Staphylococcus aureus* Network Adaptative Platform to hopefully provide a definitive answer to this question (27). In our study, 48% of patients with uncomplicated SAB and a PI received OST, with no increase in 90-day relapse or mortality compared to patients who received just parenteral therapy. It should be noted that patients in the OST group were often less severely ill than those in the IV-only group. Although we performed a multivariable analysis to mitigate this bias, the presence of unrecognized confounding factors cannot be excluded.

A key strength of the study is the concomitant evaluation of a 14-day antibiotic course and OST in patients with any PI, including heart valve and vascular prostheses that have been largely excluded from published literature. Moreover, the systematic clinical evaluation and management of all cases of bacteremia by our hospital's antibiotic stewardship team ensures a homogeneous approach, with detailed diagnostic workup and follow-up.

Limitations of the study include the retrospective and monocenter design and the relatively small number of patients included. We observed a 90-day loss-to-follow-up rate of 23%, which may have led to an underestimation of the relapse rate. However, we believe that most relapses would have been managed at the initial admitting hospital, because follow-up is generally carried out there and patients are likely to present at their nearest emergency department, which is often located at the same institution.

A further potential limitation is that we defined relapse as recurrence of bacteremia within 3 months of the index episode. This time frame was selected because reinfections generally occur beyond 3 months, whereas most relapses occur earlier (8, 28, 29). We may, therefore, have missed late relapses or erroneously classified early reinfections as relapses. However, 123 out of 188 patients had follow-up data available beyond 3 months (for a median of 15 months), with no recurrence of SAB during this period (data not shown), which makes underestimation of the relapse rate unlikely. Randomized data are needed to confirm our results and to finally answer these important questions.

## Conclusion

In a setting where full diagnostic workup and close follow-up can be ensured, the presence of a PI and OST did not seem to be associated with an increase in 90-day mortality in patients with otherwise uncomplicated SAB. Although the relapse rate was overall low, there was a non-significant trend toward a higher risk of relapse in patients with a PI. Larger-scale studies are needed.

## MATERIALS AND METHODS

We conducted a retrospective, single-center, observational study at a 726-bed tertiary care teaching hospital in Paris, France, which handles approximately 34,000 hospitalizations and 56,000 emergency consultations each year.

All patients with SAB in our institution are assessed by a member of the antibiotic stewardship team and treated according to local guidelines, which recommend 14 days of antibiotic therapy following the most recent positive blood culture, providing there is a good response to treatment and no evidence of metastatic foci. OST is recommended

after at least 5 days of IV therapy if all the following criteria are fulfilled: (i) the patient is hemodynamically stable and afebrile; (ii) follow-up daily blood cultures are sterile; (iii) endocarditis has been excluded by at least one TTE performed 5–7 days after the first positive blood culture; and (iv) source control has been achieved if indicated.

## Inclusion criteria

We extracted all non-duplicate cases of SAB between June 2015 and February 2021 from the SIRscan database (SIRscan system: I2a, Montpellier, France). All adult patients with SAB without proven or presumed metastatic foci and with a planned antibiotic therapy duration of 14 days after the last positive blood culture were included.

## Exclusion criteria

We excluded patients who died within 7 days, had a planned treatment duration of less than 7 days, had a decision to withdraw life-support therapy for ethical reasons before the end of SAB treatment, had had a previous SAB episode, died with concomitant bacteremia due to another species during SAB treatment, or were transferred to another hospital.

## Definitions

PIs included orthopedic and cardiovascular (pacemaker, implanted defibrillator, heart valve, vascular graft, and HeartMate) prostheses. Uncomplicated SAB was defined as SAB without proven or presumed metastatic osteoarticular or endovascular foci (including IE) as determined by the treating physician, regardless of the duration of bacteremia or the presence of a PI. Persistent bacteremia necessitates a more in-depth diagnostic workup to rule out metastatic foci but does not itself warrant systematic prolongation of antibiotic duration in our institution. "Presumed metastatic foci" were considered present when the clinician decided to treat for more than 14 days in the absence of proven metastatic foci. SAB with adequate source control was also considered as uncomplicated if the planned antibiotic treatment duration did not exceed 14 days after source control. Relapse was considered a recurrent positive blood culture within 3 months of the index culture (8, 28, 29) after an initial sterile follow-up blood culture, with the isolated *S. aureus* displaying the same antimicrobial susceptibility profile (except for the antibiotic used for the treatment of the index episode). SAB was considered nosocomial if the patient had been hospitalized for more than 48 h at the time of the first positive blood culture. Patients at highest risk of IE were defined as stated in the 2015 European Society of Cardiology Guidelines (30): presence of prosthetic valve, previous IE, untreated cyanotic congenital heart disease (CHD), or treated CHD with residual shunt or valve regurgitation.

The primary outcome was 90-day SAB relapse in patients with and without a PI. Secondary outcomes were 90-day mortality in patients with and without a PI and 90-day SAB relapse and mortality in patients with and without OST. If no follow-up data were available at our institution, the patient's vital status (alive or deceased) was checked on the French national public database Institut national de la statistique et des études économiques (INSEE) (in which the date of every certified death in France is recorded). If no death was recorded in this database, the patient was assumed to be alive.

## Statistical analysis

Data were recorded as absolute numbers and percentages for qualitative variables and as medians and interquartile ranges or means and ±95% confidence intervals for quantitative variables. Comparisons between groups were performed using Fisher's exact test for qualitative variables and Mann-Whitney's $U$ test for quantitative variables. A multivariable logistic regression model was used to calculate aORs for 90-day mortality, using variables associated with 90-day mortality in the univariate analysis ($P < 0.1$).

Statistical analyses were performed using Prism 10, version 10.0.2 (GraphPad Software, LLC).

## ACKNOWLEDGMENTS

No specific grant for this research was received from any funding agency in the public, commercial, or not-for-profit sectors.

The authors thank Karen Pickett for her editorial assistance.

Study design: D.B., D.L., and M.D.; data extraction: P.G.; data collection: D.B. and L.L.; data analysis and interpretation: D.B., D.L., and M.D.; original draft writing: D.B., D.L., and M.D.; critical review and editing of the draft: L.L., PG, J.L.M., and J.P.B.

## AUTHOR AFFILIATIONS

[1]Service de Microbiologie, Unité Mobile d'Infectiologie, AP-HP, Hôpital Européen Georges Pompidou, Paris, Île-de-France, France
[2]Faculté de Santé, UFR de Médecine, Université Paris Cité, Paris, Île-de-France, France
[3]Inserm UMR-S 1138, Centre de Recherche des Cordeliers, Sorbonne Université, Université Paris Cité, Paris, Île-de-France, France
[4]Genetics of Biofilms Laboratory, Université Paris Cité, Institut Pasteurt, Paris, Île-de-France, France

## AUTHOR ORCIDs

Damien Blez http://orcid.org/0000-0003-1117-9360

## AUTHOR CONTRIBUTIONS

Damien Blez, Conceptualization, Data curation, Formal analysis, Investigation, Methodology, Writing – original draft, Writing – review and editing | Luc Labarbe, Data curation, Investigation, Validation, Visualization, Writing – review and editing | Patrick Grohs, Conceptualization, Data curation, Software, Validation, Visualization | Jean-Luc Mainardi, Validation, Visualization, Writing – review and editing | Jean-Philippe Barnier, Validation, Visualization, Writing – review and editing | David Lebeaux, Conceptualization, Supervision, Validation, Visualization, Writing – original draft, Writing – review and editing | Marie Dubert, Conceptualization, Supervision, Validation, Visualization, Writing – original draft, Writing – review and editing

## ETHICS APPROVAL

This study was approved by the ethics committee of Assistance Publique–Hôpitaux de Paris (CERAPHP.5) (IRB registration: #00011928). No patient consent was required because of the retrospective, observational nature of the study. Collection and storage of data followed the French General Data Protection Regulations (RGPD–20220117103030). Written information was sent by mail to every patient alive at the time of inclusion.

## ADDITIONAL FILES

The following material is available online.

### Supplemental Material

**Supplemental material (Spectrum03337-24-s0001.docx).** Tables S1 to S4; Fig. S1.

### Open Peer Review

**PEER REVIEW HISTORY (review-history.pdf).** An accounting of the reviewer comments and feedback.

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
