## [Reviewer comments · Microbiology Spectrum]

Microbiology Spectrum

Impact of the presence of a prosthetic implant and transition to oral stepdown therapy on relapse rates and mortality in uncomplicated *Staphylococcus aureus* bacteremia treated with 14 days of antibiotics: a retrospective cohort study

Damien Blez, Luc Labarbe, Patrick Grohs, Jean-luc Mainardi, Jean-Philippe Barnier, David Lebeaux, and Marie Dubert

Corresponding Author(s): Damien Blez, Assistance Publique - Hopitaux de Paris

Review Timeline:

Submission Date:	December 19, 2024
Editorial Decision:	February 6, 2025
Revision Received:	March 28, 2025
Editorial Decision:	April 7, 2025
Revision Received:	April 23, 2025
Accepted:	April 24, 2025

Editor: Paschalis Vergidis

Reviewer(s): The reviewers have opted to remain anonymous.

Transaction Report:

DOI: <https://doi.org/10.1128/spectrum.03337-24>

Re: Spectrum03337-24 (Impact of the presence of a prosthetic implant and transition to oral stepdown therapy on relapse rates and mortality in uncomplicated Staphylococcus aureus bacteremia treated with 14 days of antibiotics: a retrospective cohort study)

Dear Dr. Damien Blez:

Thank you for the privilege of reviewing your work. Below you will find my comments, instructions from the Spectrum editorial office, and the reviewer comments.

Revision Guidelines

Sincerely,
Paschalis Vergidis
Editor
Microbiology Spectrum

Reviewer #1 (Comments for the Author):

Blez et al present a retrospective cohort study on 612 patients with S. aureus bacteremia from routine clinical practice. The selected 188 patients with "uncomplicated" SAB in who they assessed, whether prosthetic implanted (PI) devices (orthopedic AND cardiac implants) were associated with relapse and mortality. They further assessed relapse and mortality in oral treatment.

The manuscript is well written and logically composed. Nevertheless, it could be improved.

Major points:

- 1) The importance of the study is somewhat limited by the small size. 3 out of 4 patients with relapse were in the PI group, which results in an OR of 7. However, this is not significant due to its small size. This limitation needs to be clearly addressed, and the overall message should be toned down. For example, in my opinion, the study does not provide "reassuring real-world data". It rather shows that there may be more complications in the PI group, although not statistically significant.
- 2) Observational studies on oral medication are prone to indication bias, i.e. patients that are doing better tend to be put on oral medication. This should be discussed in the limitations. This may explain, why oral therapy has a better outcome. Table S4 should also list oral stepdown therapy.
- 3) Table S3 has the main outcome results and should be moved to the main manuscript.
- 4) It is not explained, how missing data is handled, esp. in the 30d and 90d mortality. Rather than reporting median follow-up time, I would suggest reporting loss-to-FU at day 30 and 90 in the outcome table.
- 5) line 137: the low treatment-failure rate is probably due to the selection of patients.
- 6) In the limitations it should be mentioned how many patients were lost-to follow-up within 3 months. This would better describe the population at risk of having an unrecognized relapse.
- 7) The definition of relapse is based on BC results. Were there any late *S. aureus* infections without a positive BC? These should also be classified as relapse.
- 8) The definition of "healthcare acquired" is not a standard definition. Rather use the term "nosocomial" which fits best to the methods used.
- 9) Bacteremic pneumonia has the worst prognosis of SAB foci. Why was it classified as "uncomplicated"?
- 10) Figure 1: please explain whether criteria in "patients excluded" overlap. This should be the case for "treatment <7d" and "death within 7d".
- 11) Table 1: please make clear that central venous catheter denotes the presence of a CVC at SAB onset. Further, more common is the term "skin soft tissue" as a focus.
- 12) Table 2 can be moved to Supplement or deleted, since it can be described in the text. It further hides that the outcome of interest is only present in 4 patients.
- 13) Table S1: why are orthopedic implants not shown?
- 14) A Kaplan-Meier curve for the secondary outcome mortality would be nice in the supplement.

Minor points:

- 1) line 48: "high-risk" is used without definition. To the reader it is unclear which definition is used.
- 2) line 109: also report number of days of therapy
- 3) line 141: There is currently no "consensus among international experts" regarding PI. Please rephrase.
- 4) line 174: are "intravenous" drug users meant?
- 5) line 217: OST was initiated after 5-7 days and required a TTE. The TTE was performed 5-7 days after the first positive BC. Is this possible? How many patients actually had a TTE before oral switch?
- 6) Table S4: add GI bleeding as cause of death in pt 3.

Reviewer #2 (Comments for the Author):

The manuscript titled "Impact of the Presence of a Prosthetic Implant and Transition to Oral Stepdown Therapy on Relapse Rates and Mortality in Uncomplicated Staphylococcus aureus Bacteremia Treated with 14 Days of Antibiotics: A Retrospective Cohort Study" is a well-structured and informative original study highlighting the outcomes of short-course antibiotic therapy in patients with Staphylococcus aureus bacteremia (SAB), particularly in those with prosthetic implants and those transitioning to oral stepdown therapy (OST).

The authors demonstrate that in a setting with rigorous diagnostic workup and close follow-up, neither the presence of a prosthetic implant (PI) nor OST significantly impacts 90-day relapse or mortality rates. Their findings provide valuable real-world data supporting the safety of short-course treatment in carefully selected patients. I appreciate their work and efforts in addressing an important clinical question. However, several key areas require further clarification due to major concerns.

Major comments

- This study includes persistent bacteremia beyond 72 hours as uncomplicated SAB, which differs from the IDSA guidelines. This discrepancy may confuse readers and should be explicitly clarified
- Exclusion criteria (1): Patients who died within seven days of SAB diagnosis were excluded. While this exclusion is understandable for focusing on treatment outcomes, it may underestimate the true mortality burden of SAB, as a significant number of patients (N=72) were excluded due to death within seven days of treatment.
- Exclusion criteria (2): Not treated/contamination (n=15): I was wondering if this label refers to contamination, as S. aureus is typically not considered contaminant pathogens in clinical practice.
- Although the inclusion criteria mentioned that patients receiving antibiotic treatment for more than 14 days were excluded, based on the results in Table S2, some patients received more than 14 days, as the IQR is 14-16 and 14-15, respectively. Please check the data.
- Are there any data available on comorbidities? These could impact treatment outcomes and should be considered.
- What is the rationale for selecting a SOFA score cutoff of {greater than or equal to}2?
- The authors mentioned '48% of patients with uncomplicated SAB who received a PI received OST, with no increase in 90-day relapse or mortality compared to patients who received just parenteral therapy.' What do you think was the main trigger for switching to oral step-down therapy? The oral step-down therapy group showed a tendency to favor better outcomes in terms of 90-day mortality, with a univariate analysis showing an odds ratio of 0.2 (0.1-0.6), although this was not statistically significant in multivariate analysis (OR 0.5, 0.2-1.4). Perhaps patients with lower SOFA scores? Readers would likely be interested in a comparison between the oral step-down therapy group and the IV therapy group.
- The authors stated in the abstract: "In a setting where a full diagnostic workup and close follow-up were ensured, the presence of a PI or OST was not associated with an increased 90-day relapse or mortality rate in patients with SAB treated with a 14-day course of antibiotics." However, this statement is misleading and should be revised for above reason.

Minor comments

- References are written in French (Month); they should be changed to English.

Response to Reviewers, "Impact of the presence of a prosthetic implant and transition to oral stepdown therapy on relapse rates and mortality in uncomplicated *Staphylococcus aureus* bacteremia treated with 14 days of antibiotics: a retrospective cohort study", by Damien Blez *et al.*

Dear Editor,

We thank you for your helpful and constructive comments on the submitted version of our manuscript and for giving us the opportunity to improve it.

All comments and suggestions have been taken into account as indicated in the provided detailed point-by-point response. All changes made in the revised version of our manuscript are highlighted in yellow and described in the authors' response.

We hope that you find our responses satisfactory, and that the manuscript is now considered acceptable for publication in **Microbiology Spectrum**.

Reviewer #1 (Comments for the Author):

Blez et al present a retrospective cohort study on 612 patients with *S. aureus* bacteremia from routine clinical practice. The selected 188 patients with "uncomplicated" SAB in who they assessed, whether prosthetic implanted (PI) devices (orthopedic AND cardiac implants) were associated with relapse and mortality. They further assessed relapse and mortality in oral treatment.

The manuscript is well written and logically composed. Nevertheless, it could be improved. We thank Reviewer #1 for his/her comments and are pleased to have the opportunity to improve our work.

Major points:

1) The importance of the study is somewhat limited by the small size. 3 out of 4 patients with relapse were in the PI group, which results in an OR of 7. However, this is not significant due to its small size. This limitation needs to be clearly addressed. and the overall message should be toned down. For example, in my opinion, the study does not provide "reassuring real-world data". It rather shows that there may be more complications in the PI group, although not statistically significant.

Authors' response: We agree with comment n°1. However, although we did not have enough patients to draw conclusions regarding comparison between groups, the 5% relapse-rate we observed in the prosthetic-group remains low compared to published data on *S. aureus* bacteremia (as detailed in the discussion section). This is particularly true for "at risk" bacteremia with prosthetic implants. Consequently, we believe that these findings remain reassuring, even though we cannot conclude that no difference exists between the two groups.

Conclusion is nonetheless modified to better fits reviewer #1 interpretation, in the abstract section lines 37-41 and in the Discussion sections, lines 207-211: **"In patients with otherwise uncomplicated SAB, presence of a prosthetic implant or OST did not seem to be associated with an increase in 90-day mortality, in a setting where full diagnostic workup and close follow-up can be ensured. Although relapse rate was overall low, there was a non-**

significant trend toward a higher risk of relapse in case of prosthetic implant. Larger-scale studies are needed.”

2) Observational studies on oral medication are prone to indication bias, i.e. patients that are doing better tend to be put on oral medication. This should be discussed in the limitations. This may explain, why oral therapy has a better outcome. Table S4 should also list oral stepdown therapy.

Authors’ response: We agree with comment n°2 and therefore added further details on this bias in the discussion section, as requested. Table S4 has been updated as requested.

“It should be noted that patients on the OST group are often less severe than IV-only group. Although we performed a multivariable analysis to mitigate this bias, the presence of unrecognized confounding factors cannot be excluded.” lines 185-187

3) Table S3 has the main outcome results and should be moved to the main manuscript.

Authors’ response: We agree with this comment, table S3 has been moved to the main manuscript, as requested (named Table 2).

4) It is not explained, how missing data is handled, esp. in the 30d and 90d mortality. Rather than reporting median follow-up time, I would suggest reporting loss-to-FU at day 30 and 90 in the outcome table.

Authors’ response: We agree with this comment and add details on how the vital status was collected:

“In the absence of follow-up within our institution, the patient’s vital status (alive or deceased) was confronted to the French national public database INSEE (that provides the date of every death collected by medical certificate in France). If no death was recorded in this database, the patient was considered alive.” line 254-257

The loss-to-FU is already indirectly reported in the Table S3 (that became Table 2) in “follow-up > 3months”: we changed the presentation to be clearer and add the loss to follow-up at day 30.

5) line 137: the low treatment-failure rate is probably due to the selection of patients.

Authors’ response: We agree with this comment. Our point, however, is that a close evaluation and follow-up are required to select the right patient for the right duration of antibiotic therapy. We changed the sentence for clarity.

“This low treatment-failure rate despite short-course treatment and presence of a PI may be explained by appropriate selection of patients through a systematic evaluation and close follow-up by the antibiotic stewards at our institution” line 134

6) In the limitations it should be mentioned how many patients were lost-to follow-up within

3 months. This would better describe the population at risk of having an unrecognized relapse.

Authors' response: We agree with this comment and added these details in the discussion section:

“We observed a 90-day loss to follow-up rate of 23%, which could lead to an underestimation of the relapse rate. However, we believe that most relapses are managed at the initial hospital, as patients are most often followed up there and because emergency departments are geographically distributed based on patients' place of residence.” line 194-197

7) The definition of relapse is based on BC results. Were there any late *S. aureus* infections without a positive BC? These should also be classified as relapse.

Authors' response: We agree with this comment; patients may theoretically relapse locally without positive BC. And if they are not treated again, they are expected to have new positive BC.

It should be noted that we excluded the at-risk organ involvements like osteoarticular or endovascular foci. We therefore probably excluded patients that were the most prone to experience a local relapse of infection without bacteremia.

In this series, none of the patients experienced a relapse without positive BC (every medical chart was checked for any new anti-staphylococcal treatment OR new diagnosis of staphylococcal infection during the 90-day follow-up).

We add in the discussion section: “While some patients may relapse without new positive blood culture, none of our patients in this series received a new anti-staphylococcal treatment during follow-up (data not shown).” (line 137-139)

8) The definition of "healthcare acquired" is not a standard definition. Rather use the term "nosocomial" which fits best to the methods used.

Authors' response: We agree with this comment and changed healthcare-acquired for nosocomial, as requested. (line 247 and Table 1)

9) Bacteremic pneumonia has the worst prognosis of SAB foci. Why was it classified as "uncomplicated"?

Authors' response: We agree with Reviewer#1 that there may be confusion about the definition of complicated vs uncomplicated bacteremia.

We stated in the definition section that SAB are considered uncomplicated if the planned duration of antibiotic therapy does not exceed 14 days, to better fit IDSA guidelines (Liu *et al.*, CID 2011) that recommends extended duration of treatment when complicated criteria is present (4 to 6 weeks compared to 2 weeks).

S. aureus pneumonia has to be treated for 1 to 3 weeks, in the same IDSA Guidelines: early mortality of these patients is indeed unlikely to be prevented by an increased duration of antibiotic therapy. Any increase in mortality due to an unduly short treatment duration must necessarily result from a recurrence of the infection.

We therefore believe that lung involvement is not a mandatory criteria to extend the duration of antibiotic therapy despite its prognosis (in accordance with IDSA guidelines), and have consequently not considered pneumonia as a “complicated feature” of bacteremia, despite its poor prognosis.

10) Figure 1: please explain whether criteria in "patients excluded" overlap. This should be the case for "treatment <7d" and "death within 7d".

Authors' response: The exclusion criteria do not overlap. Treatment < 7d means “planned treatment”.

We did not include any patient that received less than 7 days of treatment for any reasons.

We modified our manuscript to “planned treatment duration was less than seven days” in the exclusion criteria to be more explicit, as requested. (line 229)

11) Table 1: please make clear that central venous catheter denotes the presence of a CVC at SAB onset. Further, more common is the term "skin soft tissue" as a focus.

Authors' response: We agree with the first comment and added the terms “at SAB onset” in the table 1.

We also agree and changed the term to “skin and soft tissue” for the primary cutaneous focus in the Table 1.

12) Table 2 can be moved to Supplement or deleted, since it can be described in the text. It further hides that the outcome of interest is only present in 4 patients.

Authors' response: We agree with this comment. Table 2 has been moved to supplementals (named table S3), as requested.

13) Table S1: why are orthopedic implants not shown?

Authors' response: For this specific issue of diagnostic workup, we deemed it essential to specify the subgroup of cardiovascular implants, as they have direct impact on the relevance and performance of the radiologic exams performed. We thought that adding osteoarticular implants would not add any relevant information in our results.

14) A Kaplan-Meier curve for the secondary outcome mortality would be nice in the supplement.

Author's response: we added a Kaplan-Meier curve as requested in the supplement, Figure S1.

Minor points:

1) line 48: "high-risk" is used without definition. To the reader it is unclear which definition is used.

Authors' response: We agree with this comment. We considered the patients as "high-risk patients" because of the implanted prosthesis: we deleted "high risk" to be more explicit (line 45).

2) line 109: also report number of days of therapy

Authors' response: We added the duration of therapy.

"Overall, 4/188 (2%) patients relapsed after a 14-day (n=3) and 16-day (n=1) antibiotic therapy course" line 105-106

3) line 141: There is currently no "consensus among international experts" regarding PI. Please rephrase.

Authors' response: The study cited here is the most recent published data about international expert opinion and shows that most experts consider PI (even uninfected ones, with high degree of agreement) as a classifying element of complicated bacteremia. We changed the sentence to delete the term "consensus".

"Although a precise definition of complicated SAB has not been established, most experts consider PI as a component of "complicated SAB" line 140-141

4) line 174: are "intravenous" drug users meant?

Authors' response: Absolutely, we added it, thank you. line 174

5) line 217: OST was initiated after 5-7 days and required a TTE. The TTE was performed 5-7 days after the first positive BC. Is this possible? How many patients actually had a TTE before oral switch?

Authors' response: We are a center with a large cardiology technical platform, where cardiologists and antibiotic stewardship team are working in close collaboration. All our hospitalized patients can undergo TTE within 7 days following SAB diagnosis. In the rare cases where it cannot be done, the oral stepdown therapy is delayed.

6) Table S4: add GI bleeding as cause of sepsis in pt 3.

Authors' response: We added it.

Reviewer #2 (Comments for the Author):

The manuscript titled "Impact of the Presence of a Prosthetic Implant and Transition to Oral Stepdown Therapy on Relapse Rates and Mortality in Uncomplicated Staphylococcus aureus Bacteremia Treated with 14 Days of Antibiotics: A Retrospective Cohort Study" is a well-structured and informative original study highlighting the outcomes of short-course antibiotic therapy in patients with Staphylococcus aureus bacteremia (SAB), particularly in those with prosthetic implants and those transitioning to oral stepdown therapy (OST). The authors demonstrate that in a setting with rigorous diagnostic workup and close follow-up, neither the presence of a prosthetic implant (PI) nor OST significantly impacts 90-day

relapse or mortality rates. Their findings provide valuable real-world data supporting the safety of short-course treatment in carefully selected patients. I appreciate their work and efforts in addressing an important clinical question. However, several key areas require further clarification due to major concerns.

Authors' response: We thank Reviewer #2 for his comments and are pleased to answer and improve our work.

Major comments

- This study includes persistent bacteremia beyond 72 hours as uncomplicated SAB, which differs from the IDSA guidelines. This discrepancy may confuse readers and should be explicitly clarified

Authors' response: We agree with this comment and added in the definition section:

- “Regardless of the presence of a PI or the duration of bacteremia **(as recommended in our institution, persistent bacteremia implies a broader diagnostic workup to ensure the absence of metastatic foci but no systematic prolongation of antibiotic duration)**” to be more explicit. (line 238-240)

As our objective was to evaluate 14-day antibiotic therapy for patients with PI, it was relevant to include every patient in our institution that had 14 days of treatment irrespective of IDSA definition of complicated bacteremia. In any case, we believe the definition used should not impact the conclusions of our study since the inclusion criteria are explicitly detailed in the methods section.

Therefore, we had to modify the “classical” definition of complicated bacteremia since IDSA introduced the term complicated as a surrogate for “a need for more than 14 days of antibiotic therapy”

Moreover, in our study, persistent bacteremia was not associated with increased mortality (multivariable analysis) or relapse (univariate analysis). However, that was not the primary question explored but rather a relevant covariate for multivariable analysis.

- Exclusion criteria (1): Patients who died within seven days of SAB diagnosis were excluded. While this exclusion is understandable for focusing on treatment outcomes, it may underestimate the true mortality burden of SAB, as a significant number of patients (N=72) were excluded due to death within seven days of treatment.

Authors' response: We agree with this comment. We added line 135-137:

- “It should also be noted that the exclusion of patients with early mortality (before 7 days) may have led to an underestimation of overall mortality compared to published data”

We are confident, however, that it does not impact our results, since early mortality (the patient died before the end of treatment) cannot, by definition, have been impacted by the planned duration of antibiotic treatment.

•Exclusion criteria (2): Not treated/contamination (n=15): I was wondering if this label refers to contamination, as *S. aureus* is typically not considered contaminant pathogens in clinical practice.

Authors' response: This is indeed very unusual and surprising. We chose not to study these patients that were not treated, had spontaneous negative follow up blood culture and good outcome without treatment. We chose to call it contamination, as with any bacteria that is present on the skin. Most of the time, patients are treated, even with a single positive blood culture.

While debatable, this could not impact the results of our study in any way.

•Although the inclusion criteria mentioned that patients receiving antibiotic treatment for more than 14 days were excluded, based on the results in Table S2, some patients received more than 14 days, as the IQR is 14-16 and 14-15, respectively. Please check the data.

Authors' response: The data have been checked.

We chose to include all SAB with **planned duration of 14 days after the last blood culture** (line 222). The variation of duration relies on the calculation of day 1 (first negative blood culture for example, as stated in methods > inclusion criteria).

We added a footnote at the table S2 “*Some patients were treated more than 14 days, because day 1 start at the first negative blood culture” to be clearer.”

•Are there any data available on comorbidities? These could impact treatment outcomes and should be considered.

Authors' response: We agree with this comment. Unfortunately, we didn't collect other comorbidities.

•What is the rationale for selecting a SOFA score cutoff of {greater than or equal to}2?

Authors' response: SOFA score ≥ 2 is the criteria for this score to be “positive” as a predictor of poor outcome and can be used in the definition of sepsis (Seymour *et al.*, JAMA 2016). We acknowledge the shortcomings of this score (and others), as stated in the Surviving Sepsis Campaign that has been updated in 2021.

Although we could have been more precise about initial severity of patients, we believe this information to be not mandatory to evaluate a duration of antibiotic therapy, which is only marginally impacted by the initial severity of the infectious disease.

•The authors mentioned '48% of patients with uncomplicated SAB who received a PI received OST, with no increase in 90-day relapse or mortality compared to patients who received just parenteral therapy.' What do you think was the main trigger for switching to oral step-down therapy? The oral step-down therapy group showed a tendency to favor better outcomes in terms of 90-day mortality, with a univariate analysis showing an odds ratio of 0.2 (0.1-0.6), although this was not statistically significant in multivariate analysis (OR 0.5, 0.2-1.4). Perhaps patients with lower SOFA scores? Readers would likely be interested in a comparison between the oral step-down therapy group and the IV therapy group.

Authors' response: We agree with this comment. Our local guidelines recommend oral stepdown therapy every time it is possible (if there is reliable oral alternative, no

contraindication, no diagnosis uncertainty and no need for IV route for any other medical reasons).

Therefore, patients that have IV-only treatment are more likely to be severe and/or with more comorbidities. This is why we chose not to study this endpoint as a primary outcome since it is highly prone to biases, that we tried to minimize with multivariable analysis.

We believe that the retrospective study of OST in SAB warrants a separate publication, including all *S. aureus* bacteremia regardless of treatment duration.

- The authors stated in the abstract: "In a setting where a full diagnostic workup and close follow-up were ensured, the presence of a PI or OST was not associated with an increased 90-day relapse or mortality rate in patients with SAB treated with a 14-day course of antibiotics." However, this statement is misleading and should be revised for above reason.

Authors' response: We agree with this comment and added in the discussion: (Line 185-187) "It should be noted that patients on the OST group are often less severe than IV-only group. Although we performed a multivariable analysis to mitigate this bias, the presence of unrecognized confounding factors cannot be excluded."

We tempered the conclusion to "In patients with otherwise uncomplicated SAB, presence of a prosthetic implant or OST did not seem to be associated with an increase in 90-day mortality, in a setting where full diagnostic workup and close follow up can be ensured. Although relapse rate was overall low, there was a non-significant trend toward a higher risk of relapse in case of prosthetic implant. Larger-scale studies are needed." (Line 37-41 and 207-211)

Minor comments

- References are written in French (Month); they should be changed to English.

Authors' response: We corrected it, thank you.

Re: Spectrum03337-24R1 (Impact of the presence of a prosthetic implant and transition to oral stepdown therapy on relapse rates and mortality in uncomplicated Staphylococcus aureus bacteremia treated with 14 days of antibiotics: a retrospective cohort study)

Dear Dr. Damien Blez:

Thank you for the privilege of reviewing your work.

I recommend that the authors include the total duration of antibiotic treatment (median and IQR) for both study groups in the main text of the manuscript.

To enhance clarity and readability, the following sentences/phrases would benefit from revision by a medical editor.

- "In patients with otherwise uncomplicated SAB, presence of a prosthetic implant or OST did not seem to be associated with an increase in 90-day mortality, in a setting where full diagnostic workup and close follow-up can be ensured."
- "It should be noted that patients on the OST group are often less severe than IV-only group."
- "We believe that most relapses are managed at the initial hospital"
- "Persistent bacteremia implies a broader diagnostic workup to ensure the absence of metastatic foci but no systematic prolongation of antibiotic duration"
- "The patient's vital status (alive or deceased) was confronted to the French national public database"

Please return the manuscript within 30 days; if you cannot complete the modification within this time period, please contact me. If you do not wish to modify the manuscript and prefer to submit it to another journal, notify me immediately so that the manuscript may be formally withdrawn from consideration by Spectrum.

Revision Guidelines

Sincerely,

Paschalis Vergidis
Editor
Microbiology Spectrum

Dear editor,

We thank you again for reviewing our work,

All comments and suggestions have been taken into account as indicated in the detailed point-by-point response. All changes made in the revised version of our manuscript are highlighted in yellow and described in the authors' response.

We hope that you find our responses satisfactory, and that the manuscript is now considered acceptable for publication in Microbiology Spectrum.

Editor response:

I recommend that the authors include the total duration of antibiotic treatment (median and IQR) for both study groups in the main text of the manuscript.

Author response: we agree with this recommendation and have added, as requested, lines 99-101:

"The total duration of antibiotic treatment was 14 days (IQR: 14-16) for patients with a PI and 14 days (IQR: 14-15) for those without."

To enhance clarity and readability, the following sentences/phrases would benefit from revision by a medical editor.

- "In patients with otherwise uncomplicated SAB, presence of a prosthetic implant or OST did not seem to be associated with an increase in 90-day mortality, in a setting where full diagnostic workup and close follow-up can be ensured."
- "It should be noted that patients on the OST group are often less severe than IV-only group."
- "We believe that most relapses are managed at the initial hospital"
- "Persistent bacteremia implies a broader diagnostic workup to ensure the absence of metastatic foci but no systematic prolongation of antibiotic duration"
- "The patient's vital status (alive or deceased) was confronted to the French national public database"

Author response: the revised manuscript has been checked by a native English-speaking medical editor. The specific sentences mentioned by the reviewer have been altered to read:

- In a setting where full diagnostic workup and close follow up can be ensured, presence of a PI and OST did not seem to be associated with an increase in 90-day mortality in patients with otherwise uncomplicated SAB. **Line 35**
- It should be noted that patients in the OST group were often less severely ill than those in the IV-only group. **Line 186**
- However, we believe that most relapses would have been managed at the initial admitting hospital, because follow-up is generally carried out there and patients are likely to present at their nearest emergency department, which is often located at the same institution. **Line 197**
- Persistent bacteremia necessitates a more in-depth diagnostic workup to rule out metastatic foci, but does not itself warrant systematic prolongation of antibiotic duration in our institution. **Line 242**

- the patient's vital status (alive or deceased) was checked on the French national public database INSEE (in which the date of every certified death in France is recorded). **Line 258**

We have also added the following sentences:

- Prolonged antibiotic therapy may therefore not be routinely needed if infection is excluded and thorough evaluation for dissemination performed, accompanied by close clinical and biological monitoring" **Line 45**
- The authors would like to thank Karen Pickett for her editorial assistance. **Line 279**

Re: Spectrum03337-24R2 (Impact of the presence of a prosthetic implant and transition to oral stepdown therapy on relapse rates and mortality in uncomplicated Staphylococcus aureus bacteremia treated with 14 days of antibiotics: a retrospective cohort study)

Dear Dr. Damien Blez:

Your manuscript has been accepted, and I am forwarding it to the ASM production staff for publication. Your paper will first be checked to make sure all elements meet the technical requirements. ASM staff will contact you if anything needs to be revised before copyediting and production can begin. Otherwise, you will be notified when your proofs are ready to be viewed.

Sincerely,
Paschalis Vergidis
Editor
Microbiology Spectrum